# Sustainable Development: Smart Co-Operative Management Framework

**Anassaya Chawviang and Supaporn Kiattisin ***

Information Technology Management, Faculty of Engineering, Mahidol University,
Nakorn Pathom 73170, Thailand; anassaya.cha@student.mahidol.ac.th
* Correspondence: supaporn.kit@mahidol.ac.th; Tel.: +66-81-866-4207

**Abstract:** A smart co-operative refers to a co-operative that aims to apply ICT to provide better services and increase management efficiency to meet organizations' goals through the combinations of ICT technology and business. In this paper, we propose the sustainable development smart co-operative framework, which specifically applies to all types of co-operatives which use information technology in their organization, enabling transformation to improve their services, management, and governance. In addition, we discuss ICT channel creation for improving knowledge, awareness, democracy, and the participation of members, a process in which IT contributes to the accessibility of members and communication between the co-operative, members, and stakeholders. The element design of this proposed framework has considered three key principles, which are (1) smart members, (2) the smart economy, and (3) smart governance. A smart co-operative is a term used to extend the concept of a smart city into co-operative organization to promote a sustainable development approach in the co-operative sector. Therefore, the smart co-operative combines ICT, smart concepts, co-operative business aspects, business models, and innovation. The findings suggest that the smart and sustainable development co-operative framework is suitable for co-operatives, providing a comprehensive framework for value creation through the smart co-operative concept.

**Keywords:** smart concept; smart economy; smart governance; sustainable development; participation; e-administration; collaboration

## 1. Introduction

The co-operative group is one type of significant financial institution, which is normally established by a group of people working together in the same location, nation, region, or occupation. Co-operatives are created in order to provide credits, goods, and services to their members. According to the literature, at least 12% of the world's population are members of co-operatives [1]. Co-operatives are classified into eight categories based on business operations such as agriculture, fisheries, industry, craft and services, banking, insurance, retail, housing, and health care [1]. Meanwhile, disruptive technology is having a significant influence on co-operative enterprises, changing member behavior and increasing market competition. Therefore, co-operatives have to change their business processes in order to successfully serve their members. Additionally, a co-operative is an important institution that assists members in living a better life according to their vision and the co-operative's principles.

Co-operatives differ from the other business organizations in that an association of people unites voluntarily to meet its members' economic needs through a jointly owned and democratic controlled co-operative [2–4], members participate in policy setting and decision making with equal voting rights [5,6]. Co-operative values are based on concepts of self-help, self-responsibility, democracy, equality, equity, and solidarity [7,8]. Co-operatives can contribute to sustainable development and operations for members and community benefits through the policies approved by their members with democratic control [9].

Therefore, an important role of co-operatives is providing benefits [10] as well as encouraging their members' co-operative ideals [11]. Additionally, the decision-making processes and associated outcomes for members' benefits have been studied [12,13]. Although co-operatives receive income from different sources [14], they do not aim to maximize profit [2]. Co-operative objectives are created for maximizing the benefits of members and reducing fluctuations found in income and expenses such as service markets, increase in the economy, consumer goods, and improvement of the quality of life for their members [15,16]. Co-operatives can perform any form of economic activity and can be engaged in any sector of the economy. Therefore, they must comply with the relevant regulations [17], and co-operative law should be validated by all stakeholders. Presently, the business value of co-operative business has grown. However, the performance of co-operatives has been hampered by governance problems including financial scandals, the neglect of democracy, mismanagement, monopolies of administrative power, and limitations placed on the participation of members [18]. Another point is that balancing the interest of members is an emerging area of conflict because saving members want a high rate of dividend on savings whereas borrowing members want a low lending rate [16]. Hence, information technology has an important role in facilitating an increase in participation, corporate democracy, and the quality of management decisions [19].

Smart co-operatives are co-operatives that leverage information and communication technology to boost their efficiency, service quality, and management. By merging the co-operative business process model and ICT, co-operative firms have developed innovative ways to serve their members. Consequently, ICTs and other technologies play crucial roles in facilitating smart governance with the interaction and collaboration of all stakeholders in decision-making processes [20–23]. This ensures transparency and trust in co-operative management by improving collaboration, participation, and community empowerment [24]. Hence, information and communication technologies (ICTs) play a major role in the smart co-operative and smart governance framework. The framework design focuses on the transparency and efficiency of management processes. It refers to principles, concepts, policies, standards, access rights, accountabilities, and processes [25]. The mission and objectives of the organization's policy are aligned with the co-operative's principles. Additionally, the concept of a smart co-operative (SC) provides the "blueprint" for the "master plan" for the co-operative system. The SC framework creates efficiency in business processes and governance.

By leveraging technology, co-operative services will become more affordable, transparent, and efficient. Thus, good IT governance immediately increases productivity and service quality, while also enhancing organizational performance and management [26]. Therefore, IT governance encompasses organizational structures and procedures to ensure that ICT sustains and drives organizations' strategies and objectives [27]. There are two popular standards available in relation to the framework which are Control Objectives for Information and Related Technologies (COBIT) and the Information Technology Infrastructure Library (ITIL). However, both of these lack co-operative perspectives.

Both individuals' lives and the co-operative sector have been impacted by technology. Numerous types of co-operatives have profited from technological advancements in administration. Recently, the adoption of new technologies has shown an improvement in the quality of service, a reduction in the communication barrier between stakeholders, as well as cost reductions [28]. For example, cloud computing technology provides online access and control data to the user, enabling them to share information through this platform [29]. The platform is beneficial to co-operative activity and management. Improved management lowers overall costs and boosts operational efficiency. It is capable of forecasting business trends and assisting organizations in mitigating risk. In other words, modern technology has increased the accessibility of useful information.

Information technology facilitates a regulated process involving the participation of members and regulators. It identifies the risk management and implementation risk responses within its plans [30]. Consequently, it is interesting to investigate how ICTs

could help governance and create accessibility and confidence in the co-operative system. Co-operatives are expected to accomplish the goals of the co-operative sector.

Smart co-operatives are simply co-operatives that use smart technologies [31]. Many co-operatives in Thailand have implemented enterprise resource planning (ERP) to serve their members and management in tasks such as business transaction, management, and regulation. Therefore, the smart co-operative strategy is the result of technology disruptions in relation to consistency, accuracy, visibility, efficiency, and sustainability. However, the goal of improving participation through digital technology is challenging [32]. In smart co-operative strategies and processes are improved with the use of ICT, which provides an effective co-operative system and makes processes simpler. The challenge for members in accessing services and governance should be resolved through these improved processes.

Therefore, the research question in this study is "What are the factors of the smart co-operative management framework?" This research question led to building a conceptual model for smart co-operative management by evaluating it with a panel of experts and conducting a confirmatory factor analysis (CFA). In the introduction, we provided a comprehensive discussion of the sustainable development smart co-operative framework that focuses on three dimensions, smart members, smart economy, smart governance. In the Literature Review and Methodology sections, we describe how the data were collected using a semi-structured questionnaire in order to discover the characteristics that contribute to the ability of a smart co-operative to quantify expert opinions and then how the content validity index (CVI), a structured questionnaire, and confirmatory factor analysis (CFA) were used to confirm the model fit. In the Results and Discussion section, the framework for smart co-operatives, designed to enable co-operatives of all sizes and sorts, is presented.

## 2. Literature Review

### 2.1. Smart Co-Operative Concept

The smart concept was introduced in the 1990s and is already well known worldwide in the context of smart cities. In the Smart City Wheel in Cohen's model, there are six elements, as shown in Figure 1 [33]. The smart cities concept is defined as the ability of cities to enhance service quality, increase management efficiency, and meet the organizations' goals through a combination of ICT, innovations, and new technologies. The main aspects involved in making a city smart are uses of data and information for management [20]. The performance of smart cities depends on the availability and quality of knowledge communication, the level of education, and the accessibility ICTs for administration in order to achieve sustainable development [34]. Furthermore, this is critical, as organizations rely on data analytics to improve decision quality and support business decisions [35,36]. As a result, data quality has been adopted as being advantageous for the organization [37]. Smart co-operatives link business and ICT, enabling the interaction and collaboration of diverse stakeholders by ensuring that quality and integrity information will be used in a more agile and timely fashion in the decision-making process [31,38]. The smart cities methodology encompasses planning, analysis, design, decision making, and monitoring for different management levels. The smart cities concept is also related to the policy visions, mission, and objectives.

The smart co-operative framework is divided into three areas: the smart economy, smart members, and smart governance. Co-operative businesses have grown, and business processes have changed over time. New technologies provide a means of engaging with members [39], and technological systems have been bought by many co-operatives with the aim of ensuring that their business enables members to increase business efficiency, the quality of services, the quality of management, transparency, and collaboration and participation in decision-making processes [40]. Furthermore, governance is crucial in the co-operative system. In this study, we have investigated co-operative situations in which members influence the management of the co-operative by the board members and the manager and in which the board members are required to preserve the members' benefits. The performance of such co-operatives has shown various governance issues: monopolies

of power, the restriction of members' participation, corruption by management, deficiencies of transparency in the process of decision making, and the insufficiency of monitoring and control methods [18]. Improvements are greatly needed in terms of stakeholders' participation and regulation. The quality of co-operative governance has been shown to be the main issue [18]. Therefore, implementing ICT could create good governance and a sustainable organization. As a result, we have developed the smart co-operative governance framework to ensure that an organization can fulfill its regulatory, legal, and management objectives.

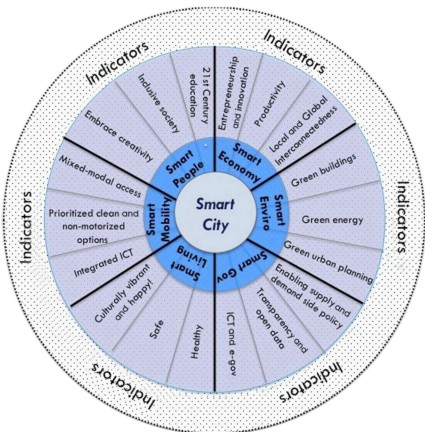

**Figure 1.** The Smart Cities Wheel proposed by B. Cohen.

2.1.1. The Smart Member

Smart members are the people making up the smart co-operative. The smart member concept, which is an extension of the smart people concept, focuses on technology, the level of qualification and education, and the quality of social interactions related to the integration of citizens as the main driver of organization with a high quality of life [31,41]. Additionally, smart people are concerned with the development of digital skills to improve the participation and efficient use of services such as e-skills in using ICT applications and services in a city [41]. The members play an important role in driving a smart co-operative to success. The co-operative's members influence the co-operative management because they have many roles in supporting the co-operative as patrons, investors, owners, and members [42,43]. Co-operatives are controlled by board members who are the representatives of the members and who voluntarily work for the co-operative [18]. A member has only one vote disregarding the number of shares owned [6,16,44]. The results of several empirical studies have shown that there are performance problems and various governance issues found in co-operatives such as financial scandals, failure of democracy, and a lack of participation by members in their boards' work [18]. Members are also rarely involved in the election of board members, and the members participate insufficiently in monitoring and controlling processes [18]. Participation is a process in which members of the public or stakeholders are involved in making decisions that impact themselves [45]. Co-operatives assign the right to make management decisions to their board of directors, but the real authority is delegated through voting by the members [44]. Therefore, co-operatives should have board performance evaluation mechanisms and encourage members to participate and monitor the management of the co-operative. Consequently, the participation of the members can improve decision-making processes [27,46].

The smart member concept is related to the degree of education and qualification of the members and the quality of co-operative interactions [41]. Knowledge is the most valuable asset of members in the organization; thus, improving knowledge is one of the principles that can guide and ensure that co-operatives achieve effective sustainable development [7,9]. Additionally, a co-operative should implement training for its members to become more financially efficient [47]. Hence, the organization should educate its members about laws,

policies, systems, and services [48]. Through investment in education and information for co-operatives members could promote a better understanding of the benefits available to the members [9]. Moreover, training is important for educational activities in co-operatives to increase productivity and efficiency, representing a part of the educational activities realized by the co-operative not only in terms of employee training but also in the training of members, managers, and the board of directors. Education and training are rooted in co-operative principles [7,49].

Furthermore, members should engage and communicate with each other not only to exchange information with the others regarding services but also to provide information to policy-makers [47]. Technology provides opportunities for increasing awareness and enabling stakeholder involvement, playing a crucial role in improved efficiency and communicating the effectiveness of policies and the services delivered [20], and ICT is the main means of education, creating accessibility and communication between the co-operative, member, and stakeholders [20,49]. Digital technologies mediate interactions between members in the co-operative, and members can act as e-participants in participation processes for potential cost saving [32,39,50–52]. ICT can also be used to explore and fulfill members' needs and provide meeting services and products [53]. The smart member concept is the result of integrating new technologies for knowledge management by creating channels for improving knowledge, awareness, democracy, communication, and the participation of members [20,54].

### 2.1.2. The Smart Economy Concept

The smart economy concept includes economic competitiveness and innovation which involves the application of ICT to economic growth and establishment and promotes innovation, contributing to functional improvements and improvements in competitiveness and enhancing efficiency [41,55]. Co-operatives can be categorized by their purpose of providing goods and services to their members. Currently, digital technology plays a significant role in co-operative businesses. Co-operative organizations intend to develop their digital capabilities to contribute business value and competitive advantage through technological innovation [6]. Innovation enables cost reduction, administration expenditure reductions, and the improvement of business processes [56,57]. It creates success in re-designing business models in alignment with organization objectives, business processes, technology, and competencies through the use of technology and innovation [58]. Technology developments in e-services have expanded competitive advantages [59], with an influence on customer experience [60] and improvement in customer interaction through new process designs [61]. Therefore, business model innovations or the digital business model use digital value drivers and digital technology, integrating technology and business to generate business value by using innovation capability to create the best offers for their customers, which later helps to improve customer experience and increase competitiveness [58,62]. As a result, digital transformation affects the business architecture in co-operatives that focus on the important issue of the sustainable development of their ability to deliver value to their members [15]. According to the co-operative concept, businesses deliver value to their members, and IT-based tools can result in participation in the co-creation of value, a culture that encourages smart operations, smart service, and smart administration [62–64].

The smart concept not only refers to technologies but also focuses on establishing policies that support important management processes [47,53]. Therefore, the challenges of the smart economy need to be systematically developed by applying the methodology for systematic digital business modeling in relation to business and IT understanding. The smart economy framework is an innovative business model for the creation, proposition, and delivery of value to its members and stakeholders by using technology in business operations. Additionally, the smart economy uses the capacity of innovation and technology for collaboration and participation in decision making by using e-administration connected with organizational processes to create business value and improve services and productivity [41,57,65]. In addition, knowledge and skills are essential for boost-

ing co-operative performance; thus, board members and staff must obtain specialized knowledge and abilities [18,45]. Therefore, collaboration among members can improve decision-making processes [27] Furthermore, a data-driven approach to organizational decisions can create value from data and enhance competitiveness and business administration [66]. Data are a key element in digital systems and are used for analysis, planning, and prediction [58]. Thus, the adoption of ICT creates more information for decision making in risk management, enabling one to avoid repeating the same mistakes [67] and providing a great diversity of services that directly affect the quality of life of the citizens [57]. Therefore, ICT is an important factor in this framework, required to successfully achieve the co-operative objectives. Participation in the co-creation value has been considered in prior studies [68]. The smart economy concept leads to a smart business that uses ICT to create smart business processes for creating beneficial economic outcomes and productivity improvements [68].

### 2.1.3. The Smart Governance Concept

Smart governance is a key component of the smart city concept [27], referring to good management with transparency and participatory democratic decision making supported by ICT [41]. Smart governance is related to the participation and engagement of various stakeholders in decision-making processes, the transparency of governance systems, and public services [69,70], and the use of ICT to mediate interactions in governance is referred to as e-governance. In a smart city, ICT empowers people and keeps decision making and execution transparent. Smart governance consists of several key elements that relate to the usage of new channels for "e-governance and e-democracy" as portals for member participation and e-voting [53,71,72], using ICT to enable and improve participation and services for citizens, to support the democratic decision-making processes, and to enhance transparency in governance [32,65,73]. E-governance is a tool which encourages citizen-centric and citizen-driven concepts in a smart city [74]. Many studies have focused on the smart governance concept in the context of smart cities, requiring administrators to make strategic, effective, credible, and achievable decisions [75].

Governance is an essential issue for management in an organization [76]. However, there are several studies on smart governance that have not studied the co-operative context. The co-operative is an organization founded by a group of people who are the co-operative's members and in which the principles of democracy are used to run the business according to the co-operative's objective. According to the literature, co-operatives work for the sustainable development of their societies through policies approved by their members [66]. The key to successful corporate governance in a co-operative can be found in an effective social system with feelings of ownership that contribute towards success which engage and motivate members, representatives, and board members [11]. However, earlier research and expert perspectives on co-operative performance have pointed to several governance issues, including accounting crises and democratic failures [18].

At the present time, corporate governance is an important driver that significantly influences high-tech enterprise innovation, enhancing the accuracy and effectiveness of decision making [62] and encouraging the responsible provision of financial information [77]. Hence, encouraging disclosure by the organization and adhering to the best corporate governance practices influences co-operatives' financial efficiency [47,78]. Information disclosure is a governance mechanism, especially in regard to financial statement information, that is necessary for enhancing good governance and improving transparency [79–81]. In addition, auditing is a mechanism of monitoring and governance based on controlling the quality of accounting information [82]. Financial statements are commonly used to improve governance and promote effective management [5]. Moreover, creating efficient internal controls is a crucial control mechanism; these include duty segregation, auditing, and a control committee, with internal control considering various risks that an organization faces and providing an effective management mechanism to prevent the occurrence of all risk cases [83]. This may help to reduce fraudulent risk [84].

ICT makes an organization smart by using data and information in facilitating sustainable management, which enhances and efficiently delivers services to stakeholders [85]. ICT has an important role in supporting information sharing, integration, and monitoring [22,63,76]. Therefore, the role of technology has become a key part of enabling new governance processes [86]. The findings of empirical studies showed that IT is a crucial factor in supporting, sharing, and integrating information between organizations and stakeholders. Furthermore, ICT is a tool that enables participation and collaboration for supporting governance mechanisms [50,63]. It can encourage collaborative governance and promote participation and engagement in an organization [77]. Integrating knowledge from diverse actors as part of ICT development and implementation contributes to an environment of collaboration within an organization [76,87].

Collaboration and participation are the main parts of the governance mechanism. Collaboration means sharing responsibility and authority for decisions related to executing policies and action plans [88]. Smart governance focuses on the concepts of collaboration and participation in the smart city, which refers to participation in decision making. Moreover, innovation enables data-driven decisions and improves the effectiveness of policy making and the integration of information [89]. Additionally, knowledge sharing to maximize the value of information reduces duplication in data collection; thus, it becomes more efficient and transparent and could deliver better quality of services. Citizen-centric e-governance focuses on ICT to enhance the ability of citizens to democratically engage with political discourse and decision making and influence changes in public policy [90,91]; e-governance in particular plays a key role in engaging citizens in these initiatives and keeping the decision-making processes transparent supported by ICT [41].

In addition, another key role of smart governance is IT governance, as an umbrella term revealing the use of technologies to achieve participation. Hence, ICT governance processes are necessary for effectiveness and efficiency [52,92]. IT-based tools can lead to expansion and participation, transforming democratic and management processes, which relate to the use of IT in the scope of administrative governance [22,27,62,76]. The main aspect of making a smart co-operative is the use of information for management [20].

Furthermore, data analytics are important in co-operatives in relation to value creation and business change [93], both of which support business decisions as well as other organizations. ICT capability is relevant to data quality management [37]. Consequently, data quality is critical for effective decision making in organizations [36,93]. Data quality consists of accuracy, timeliness, completeness, and consistency [94,95], which are indicators of the quality management of corporate data [37]. Therefore, it is essential to be aware of the data quality, with the outcomes of the expected benefits of increasing transparency [96–98]. Hence, the adoption of data quality is advantageous for the co-operative because it is a mechanism that is based on standardization designed to generate value for every kind of organization [93]. Therefore, data governance is a key element of data quality procedures to ensure that the data are of high quality and reliable [99]. Data governance specifies the rights and accountabilities in relation to data decision making in an organization [96,100], including the implementation of management information to insure proper data sharing [100,101], data policies, and standards. Data governance establishes authority and control over the management of data, aiming to increase the value of data and minimize data-related costs and risks [94,102], including assigning decision-related rights and duties for preventing unauthorized access [103]. Risks associated with data leaks or compromised methods of data collecting are frequently abused for financial benefit. Therefore, the data need to be protected in the case of secure and private information [104]. However, data governance has to align with the goals of the organization as a whole [105]. The important issues are protecting shareholder rights and information, disclosure, transparency, and paying serious attention to adherence to regulatory norms concerning board quality [106].

## 2.2. Business Model and Business Process Management (BPM)

The business model part of the framework creates success through re-designing a business model which aligns with organization objectives, business processes, technology, and competencies through the use of technology and innovation. This business model innovatively integrates technology and business to generate business value and create the best offer for customers. The framework includes a digital business model and service design methodologies which focus on value creation, value proposition, and value delivery for both customers and other stakeholders [57,107–109]. The best value-creating proposition for balance should consider all of the stakeholders' needs and the importance of a competitive advantage [110]. On-line service is a channel that uses ICT-enabled services and improves efficiency and quality of services in order to fulfill customer needs and increase quality of life [53,107]. Firstly, value co-creation could lead to the achievement of competitive advantage, which is necessary to create innovative services that facilitate interactions between service providers and customers to meet the needs of the customers [110–112]. This concept expresses the notion that firstly users and customers can impact providers by playing two roles as value creators and as consumers [113]. Secondly, the value propositions and value delivery are created by the provider, with an understanding of the customer's experience established through an interactive process to establish value and customer needs to deliver services to customers [114]. Business process management (BPM) is an organizational discipline in which an organization analyzes its current state and identifies areas of improvement to create a more efficient and effective organization. BPM needs to be aligned with the overall strategy of an organization through continuous methods, facilitating process modeling or process analysis and process improvement techniques as well as IT solutions that increase effective actions, enabling improved business performance, governance, and transparency.

## 2.3. Control Objectives for Information and Related Technologies (COBIT) Business Model

COBIT is a framework of IT management developed by the ISACA and is well known as an IT governance framework [27] helping organizations to meet challenging business goals in the areas of information management and governance. The framework involves organizing and implementing strategies, regulatory compliance, risk management, and alignment with an IT strategy. It is a framework for governance and IT management, with the objective of helping organizations create value from IT [115]. The framework can be used as an umbrella to integrate all the processes in the organization. In general, the framework is designed to make businesses more flexible and to customize a strategic IT governance approach, which (i) improves organization effectiveness and efficiency, (ii) increases competitive advantage, and (iii) maximizes profitability [115]. Furthermore, COBIT links business with IT by creating processes between IT silos and outside departments as well as aligning business goals with IT goals. The difference between COBIT and other frameworks is that it focuses on security, risk management, and governance information [116]. Security is the main issue for organizations that deal with private information and confidential data [117]. In addition, many organizations are concerned about individual privacy and personal data due to sensitive personal information being the main issue for users [118]. Therefore, it is important to establish privacy policies and organizational regulations to reduce individual privacy concerns [119]. Therefore, COBIT is a framework for managing and governing enterprise information and technology across the organization. It provides control over information technology and organizes the framework of IT-related processes [120]. COBIT 2019 is designed for the development of a governance strategy and provides the flexibility to suitably customize this governance strategy. The components of the sustainable governance system are processes, policies and procedures, organizational structures, information flows, skills, infrastructure, and culture and behaviors. It can also help to align IT with a business goal and monitor the business's performance in the organization, especially in terms of security compliance, information security, and risk management.

### 2.4. Information Technology Infrastructure Library (ITIL)

ITIL is a framework used to ensure improved ROI on IT investments, providing a guideline for implementation, reducing risks, and optimizing costs for the growth and success of an organization [121]. It focuses on aligning IT services and business needs. It involves four key concepts—the service value system, the guiding principles, the four dimensions of service management, and practices. Practices are crucial to the effective operation and value creation activities of the service value chain [122]. ITIL v3 is based on service strategy, service design, service transition, service operation, and continual service improvement. A relatively straightforward transition from the previous approach to ITIL 4 can be obtained by leveraging the service value chain and, thus, realizing the benefits it can bring [123]. ITIL4 focuses on novel IT organizations in the areas of digital transformation, customer experience, and the drive for better service [123]. Moreover, the framework enables value co-creation. This is one of the keys of the service value system in which all components and activities in the enterprise are working as a system. The service value chain is the central component of the service value system, governance, practices, guiding principles, and continual improvement. Governance sets directions and controls mechanisms. Practices are the sets of organizational resources to perform work and accomplish objectives. The guiding principles are recommendations that guide the organization in all circumstances and apply to every activity of the service value chain to ensure that stakeholders' expectations are met.

### 2.5. Proposed Framework

The smart co-operative concept is an expansion of the smart cities concept. The smart co-operative is defined as the ability of co-operatives to provide better services and increase management efficiency to meet the organizations' goals through the combinations of ICT, smart concepts, the co-operative business aspect, business models, and innovation. The smart co-operative objective serves business management and governs an organization to meet the organization's goals by increasing the transparency of management and providing cost efficiency. The smart co-operative procedure aims to link business and ICT together. The smart co-operative framework is divided into three dimensions which are smart members, the smart economy, and smart governance. ICT facilitates all these areas by connecting the business procedure in each area to promote efficiency and effectiveness in co-operative services, management, and governance.

The first component of the smart co-operative is the smart member. Empirical studies have revealed that the participation, knowledge, and communication of members are the critical factors in co-operative organization because the members, as the founders of the co-operative, have an impact on the smart co-operative's success and influence the co-operative's management, as each member has four roles in the co-operative: as a customer, investor, owner, and member [43]. Therefore, the smart member concept involves the participation, knowledge, and communication of each member, which could help in the co-operative's sustainable development. In other words, the framework uses ICT to promote member participation, member education, and communication. In the smart co-operative, knowledge is not only important to the board management but also to the members. The smart co-operative should ensure that its members understand laws, policies, co-operative systems, and services. Furthermore, members should contact and communicate with each other not only to exchange information regarding services but also to provide their information to policy-makers.

The smart economy is the second component of the smart co-operative. Nowadays, ICT has become an essential part of co-operative businesses. Co-operative organizations develop their business process by using digital capabilities to contribute to business value in order to be successful. ICTs enable the co-operative to explore and fulfill its members' needs by providing services such as a meeting portfolio, which provides comprehensive services to help the co-operative optimize services and products and to re-design business processes aligning with the co-operative's objectives. The smart co-operative combines technology

with a co-operative business process in order to generate business value, which focuses on value co-creation, value proposition, and value delivery to co-operative members. However, creating a value proposition should consider balancing the stakeholders' needs and competitive advantage. It also enables effectiveness and efficiency for the co-operative. The need for a smart economy is emphasized not only in relation to policy and processes re-design but also in relation to technologies. Thus, the smart economy component is an innovative business model for collaboration, value co-creation, value proposition, and the delivery of value to the members and stakeholders through the use of technology. A smart economy aims to improve services and co-operative management. Additionally, collaboration and co-creation by members can improve decision making and administration in the business processes. The smart economy involves considering the adoption of ICT to improve business processes and provide information to support co-operative business processes in order to achieve the co-operative's objectives.

Finally, smart governance is the third component of the smart co-operative. The smart co-operative is governed via democratic control as a decision-making process that follows the regulation based on co-operative principles. There are various stakeholders involved in the governance process, such as members, regulators, and board members. Additionally, monitoring and evaluation are mechanisms of management control in a co-operative, monitoring the performance of the co-operative in order to improve effectiveness and efficiency. ICTs play a crucial role as the main mediated governance, as e-governance enables access for stakeholders to participate in the governance process. IT governance is essential for data quality control, security, and risk management. Access control is critical in terms of information governance, especially in relation to personal data because sensitive personal information is the main issue for users. On the other hand, information disclosure is a governance mechanism, especially the financial statement—information that is necessary for enhancing good governance and improving transparency. In addition, ICT is the main issue relating to responsibility, openness, transparency, access to data, and the regulations that influence governance [71]. Therefore, smart governance is a process that could help with co-operative effectiveness, efficiency, and transparency.

## 3. Methodology

The research methodology of the smart co-operative framework is presented in Figure 2. The smart co-operative framework is a digital co-operative system designed to fill the gap in the delivery of a conceptual framework for co-operative organization in relation to sustainable development with smart concepts, business models, and guidelines or IT standards to contribute towards information technology capabilities for smart co-operatives. Therefore, this paper emphasizes the importance of identifying the proper research methodology that is suitable to obtain the expected result, according to the framework proposed, and which should be able to support the assumptions of the framework design.

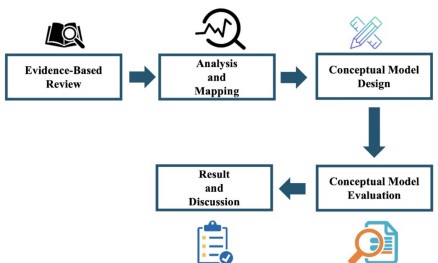

**Figure 2.** Research methodology.

### 3.1. Evidence-Based Review

The key result of this study is the smart co-operative framework for digital co-operative systems to fill the gap in the delivery of a conceptual framework for co-operative organizations in relation to sustainable development with smart concepts. The framework

consists of smart concepts, business models, and guidelines or IT standards to contribute to the information technology capabilities of smart co-operatives. Therefore, this paper emphasizes identifying the proper research methodology that is suitable for the expected result and which, according to the framework proposed, should be able to support the assumptions of the framework design. Hence, the proposed research methodology is shown in Figure 2.

This process requires an understanding of the available concepts (the smart cities concept, as well as the concepts of smart members, smart economy, and smart governance), related with the frameworks under diversified contexts, in many countries, and in relation to other specific frameworks. This means studying the capability of the adopting IT in the context of smart city initiatives which governments and their stakeholders have adopted in order to improve their services and management. In addition, we undertook a review of the literature related to the smart city concept that focused on any challenges or barriers that would affect the achievement of the adoption of this framework in our work. Additionally, it was necessary to obtain an understanding of co-operatives' goals, business processes, current problems, and issues because these influence the design and development of the framework. Therefore, the goal of this study was to conduct and improve on the smart city framework in terms of its suitability to the smart co-operative context, with the aim of contributing to sustainable development in the co-operative sector through the use of information technology to provide better quality services and management, as well as good governance with better participation and collaboration due to the principles and standardization provided through information technology management. The key topics that we reviewed became the key factors in this research, and our findings are described in Table 1.

*3.2. Analysis and Mapping*

The analysis and mapping step began with adaptability over the organization management. This was followed by mapping the expectation of the practical outcome of the framework (in a scenario with co-operative service improvements and management) into the smart co-operative framework. We then identified the factors and classified them into three groups: smart members (knowledge, participation, and communication), the smart economy (collaboration, value proposition, and value co-creation), and smart governance (democracy, regulation, monitoring and evaluation, and corporate governance), as shown in Table 2. Hence, the goal of the analysis was to identify the concepts, principles, and standards required to design and develop a smart co-operative architecture framework which focuses on using ICT in business processes, management processes, and IT governance according to the strong reference IT standard frameworks (COBIT and ITIL) for implementations in governance and management both of the private and public sector. Consequently, we would expect the results of this analysis to lead to opportunities for proposing sustainable development frameworks for the co-operative sector, especially in developing countries.

**Table 1.** The selected factors based on our literature review.

| Variables | Reference |
|---|---|
| **Knowledge** | Garcia Alonso and Lippez-De Castro (2016), Koltay (2016), Meijer and Bolívar (2016), Vázquez et al. (2018), Alves, Ferreira, and Araújo (2019), Kirimtat et al. (2020), Yobe, Ferrer, and Mudhara (2020), Piasecki (2021). |
| **Participation** | Mishra (2013), Jho and Song (2015), Rose, Qiao Liang, George Hendrikse, Zuhui Huang (2015), Persson, and Heeager (2015), Garcia Alonso and Lippez-De Castro (2016), Meijer and Bolívar (2016), Chareonwongsak (2017), Pereira et al. (2018), Reed et al. (2018), Alves, Ferreira, and Araújo (2019), Anggraeni, Gupta, and Verrest (2019), Tomor et al. (2019), Stratu-Strelet et al. (2021). |
| **Communication** | Caragliu, del Bo, and Nijkamp (2011), Falconer and Mitchell (2012), Lombardi et al. (2012), Mishra (2013), Garcia Alonso and Lippez-De Castro (2016), Meijer and Bolívar (2016), Alves, Ferreira, and Araújo (2019). |
| **Collaboration** | Harrison et al. (2012), Hans J Scholl and Scholl (2014), Jho and Song (2015), Rose, Persson, and Heeager (2015), Garcia Alonso and Lippez-De Castro (2016), Meijer and Bolívar (2016), Rodríguez Bolívar (2018), Alves, Ferreira, and Araújo (2019), McKillop et al. (2020), Oliveira, Oliver, and Ramalhinho (2020) |
| **Value Proposition** | Bocken et al. (2013), Mishra (2013), M. S. Carvalho (2015), Pérez-González and Díaz-Díaz (2015), Koltay (2016), Oswald and Kleinemeier (2016), Baldassarre et al. (2017), Gil-Garcia, Zhang, and Puron-Cid (2016), Larivière et al. (2017), Luo, Guo, and Jia (2017), Payne, Frow, and Eggert (2017), Juga and Juntunen (2018), Vázquez et al. (2018), Y. Lin (2018), Ciuchita, Mahr, and Odekerken-schröder (2019), Loukis, Janssen, and Mintchev (2019), R. Lin et al. (2020), Sheng, Amankwah-Amoah, and Wang (2017). |
| **Value Co-Creation** | Bocken et al. (2013), Charalabidis and Koussouris (2014), Voorberg, Bekkers, and Tummers (2015), Díaz-Díaz and Pérez-González (2016), Garcia Alonso and Lippez-De Castro (2016), McKillop et al. (2020), Osborne, Radnor, and Strokosch (2016), Oswald and Kleinemeier (2016), Baldassarre et al. (2017), Payne, Frow, and Eggert (2017), Cronemberger and Gil-Garcia (2019), Hamidi, Gharneh, and Khajeheian (2020), QAMAR, AHMAD, and FAROOQ (2020). |
| **Democracy** | Borgstro (2013), Mishra (2013), Hans J Scholl and Scholl (2014), Chatfield, Reddick, and Brajawidagda (2015), Chatfield, Reddick, and Reddick, Chatfield, and Jaramillo (2015), Lin, Zhang, and Geertman (2015), Qiao Liang, George Hendrikse, Zuhui Huang (2015), Díaz-Díaz and Pérez-González (2016), Garcia Alonso and Lippez-De Castro (2016), Koltay (2016), Meijer and Bolívar (2016), Ruostesaari and Troberg (2016), Viale Pereira et al. (2017), Pereira et al. (2018), Rodríguez Bolívar (2018), Alves, Ferreira, and Araújo (2019) Y. Lin (2018), Blanc (2020), Oliveira, Oliver, and Ramalhinho (2020), Reis et al. (2020), Stratu-Strelet et al.(2021) |
| **Regulation** | Kurimoto (2013), Garcia Alonso and Lippez-De Castro (2016), Meijer and Bolívar (2016). |
| **Monitoring and Evaluation** | Charalabidis and Koussouris (2014), Garcia Alonso and Lippez-De Castro (2016), McKillop et al. (2020), Chareonwongsak (2017), Viale Pereira et al. (2017) Rodríguez Bolívar (2018), Manita et al. (2020). |
| **Corporate Governance** | Borgstro (2013), Chang et al. (2014), Kwon, Lee, and Shin (2014), Yeganeh, Sadiq, and Sharaf (2014)Arnold et al. (2015), Chatfield, Reddick, and Brajawidagda (2015), Reddick, Chatfield, and Jaramillo (2015), Alonso and Lippez-De Castro (2016), Hans Jochen Scholl and Alawadhi (2016), Gil-Garcia, Zhang, and Puron-Cid (2016), Mathuva (2016), Meijer and Bolívar (2016), Ghasemaghaei, Ebrahimi, and Hassanein (2018), Hon and Millard (2018), van den Broek and van Veenstra (2018), Abraham, Schneider, and Brocke (2019), Ferramosca (2019), Saeidi et al. (2019), Wang et al. (2019), Al-Ruithe and Benkhelifa (2020), Chang, Chang, and Liao (2020), Janssen et al. (2020), Reis et al. (2020), R. Lin et al. (2020), Verdegay and Rodríguez (2020),Yobe, Ferrer, and Mudhara (2020). |

**Table 2.** The key topics identified as key factors based on the literature review.

| Authors/Research Article/ Standard/Framework/Concept | Smart Member | | | Smart Economy | | | Smart Governance | | | |
|---|---|---|---|---|---|---|---|---|---|---|
| | Knowledge | Participation | Communication | Collaboration | Value Proposition | Value Creation | Democracy | Regulation | Monitoring | Corporate Governance |
| Stratu-Strelet et al. (2021) | | X | | | | | X | | | |
| McKillop et al. (2020) | | | | X | | X | X | | X | |
| Oliveira, Oliver, and Ramalhinho (2020) | | X | | X | | | X | | | |
| R. Lin et al. (2020) | | | | | X | | | X | | X |
| Yobe, Ferrer, and Mudhara (2020) | X | | | | | | | X | | X |
| Alves, Ferreira, and Araújo (2019) | X | X | X | X | | | X | | | |
| Pereira et al. (2018) | | X | | X | | | X | | | |
| Reed et al. (2018) | | X | | | | | | X | | X |
| Rodríguez Bolívar (2018) | | | | | | | | X | X | |
| Vázquez et al. (2018) | X | | | | X | | | | | |
| Y. Lin (2018) | | | | | X | | X | | | |
| Baldassarre et al. (2017) | | | | | X | X | | | | |
| Chareonwongsak (2017) | | X | | X | | | | X | X | |
| Payne, Frow, and Eggert (2017) | | | | | X | X | | | | |
| Viale Pereira et al. (2017) | | | | | | | X | | X | |
| Díaz-Díaz and Pérez-González (2016) | | | | | | X | X | | | |
| Garcia Alonso and Lippez-De Castro (2016) | X | X | X | X | | | X | X | X | |
| Koltay (2016) | X | | | | X | | | X | | X |
| Meijer and Bolívar (2016) | X | X | X | X | | | X | X | | X |
| Chatfield, Reddick, and Brajawidagda (2015) | | | | | | | X | | | X |
| Jho and Song (2015) | | X | | X | | | X | | | |
| Charalabidis and Koussouris (2014) | | | | | | X | | | X | |
| Hans J Scholl and Scholl (2014) | | | | X | | | X | | | |
| Bocken et al. (2013) | | | | | X | X | | X | | |
| Borgstro (2013) | | | | | | | | X | | X |
| Mishra (2013) | | X | X | | X | | X | | | |

Following the identification of the elements, an expert panel discussion was held to determine the critical success factors for smart co-operatives. This stage was led by BA specialists with considerable expertise in co-operative and technical management. The evaluation approach was based on content validity using a semi-structured questionnaire completed by an expert panel. The significance of the factors was determined by nine subject area experts who assessed the factors' relevance. The scale was constructed as follows: "4 = extremely relevant", "3 = quite relevant", "2 = somewhat relevant", and "1 = not relevant". The CVI was calculated as the number of experts who provide a rating of 3 or 4 divided by the total number of experts who provide an agreement rating [124]. The CVI must be more than 0.8 to be considered acceptable. All nine experts in this study assessed all aspects to be significant. Thus, all ten elements of the CVI were within an acceptable range based on the relevant evaluations of nine experts who assigned a score of 3 or 4 to each of the ten factors.

### 3.3. Conceptual Model Design

Referring to the previous step, the most important component was the smart concept, followed by the business model and IT standard. These were the results from the business alignment with IT and governance which can create value. Fundamental conceptual frameworks have integrated governance in the business layer. The business is as focused as the strategy and service. The principles of IT and non-IT must provide the basic requirements of co-operative systems. Standards and principles are interesting in regard to e-services, e-commerce, and e-governance in the smart co-operative system because

these are the core factors in the creation of effective management. The smart co-operative could thereby increase management efficiency, governance, participation, collaboration, and monitoring mechanisms.

According to the conceptual model mapping, a researcher can make a pre-conceptual model. The conceptual model should consider the business services and business management, governance, standards, and technology. The researchers integrated smart concepts, the business framework, and information technology management frameworks including COBIT and ITIL because smart co-operatives need to solve management and governance issues. Business alignment is the basis of IT governance.

The concerns of the conceptual model are ensuring the efficiency value of co-operative services and management. The smart economy is validated by business values and management efficiency. Smart governance is validated by the co-operative principle and governance issues. Smart members enable the smart co-operative to meet its goal. The information system architecture covers the establishment of all parts of the model, especially the data and core applications, which support the driving of processes and functions to achieve the co-operative's goals. The relevant IT standards can be used to govern and manage the co-operative's business processes. This is shown as a conceptual model in Figure 3.

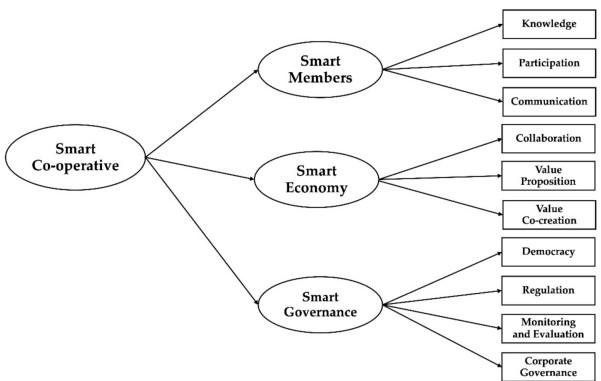

**Figure 3.** Smart co-operative conceptual model.

*3.4. Conceptual Model Validation*

In our quantitative studies, we examined the proposed conceptual framework by means of a survey among co-operative stakeholders through online questionnaires. The structure questionnaires were conducted depending on the literature review as shown in Table 1. There were 38 items measuring ten constructs on a five-point Likert scale ranging from strongly disagree to strongly agree. The validity test was an index of item objective congruence (IOC) with expert investigations, which was used for content validity [30,123–126], and the alpha coefficient method, or Cronbach's alpha, was selected for reliability analysis with pilot tests used for content reliability [30,127,128]. The questionnaires were refined before being distributed to co-operative stakeholders. The data analysis methods, which were structural equation modelling (SEM) and confirmatory factor analysis (CFA), were used to analyze on the survey data. In this study, the primary data were collected using online questionnaire surveys from co-operative stakeholders, such as members, board members, managers, staff, and regulators. The sampling units were co-operative stakeholders in Thailand through the online survey distributed to co-operative stakeholders in all regions in the country. The sample size was determined based on types of statistical power analyses and effect size [129]; therefore, this study calculated the sample size through G*Power software, version 3.1.9.4. G*Power calculated sample size with input types of statistics test and parameters, which consisted of alpha ($\alpha$), power, degree of freedom (df), and effect size. In this study we use Chi-square test, $\alpha = 0.05$, power = 0.95, df = 38, and effect size = 0.3 based on Cohen's suggestion [130]. Finally, 431 samples were

collected for this study, which corresponds to the statistical SEM technique that requires a sample size of at least 200 [131].

### 3.4.1. Reliability Analysis

To assess the reliability of the research model constructed in this study, Cronbach's alpha-based test was used to identify the reliability of the constructs. In the literature, some authors suggested that Cronbach's alpha could be acceptable if it is 0.6 or above and proposed four degrees of reliability: excellent (0.90 and above), high (0.70 to 0.90), moderate (0.50 to 0.70), and low (0.50 and below) [99,125]. In this study, proving the reliability of the construct items of the smart co-operative research model was necessary for testing. A total of 38 items measuring ten constructs were assessed for reliability; these items and constructs are shown in Table 3, which presents a summary of the reliability analysis results in this study. The results show the following Cronbach's alpha values: for the three items measuring knowledge (KL)—0.851; for the three items measuring participation (PT)—0.919; for the three items measuring communication (CC)—0.907; for the four items measuring collaboration (CL)—0.908; for the five items measuring value proposition (VP)—0.954; for the three items measuring value co-creation (VC)—0.939; for the three items measuring democracy (DC)—0.697; for the four items measuring regulation (RL)—0.827; for the five items measuring monitoring and evaluation (ME)—0.934; and for the three items measuring corporate governance (CG)—0.961.

**Table 3.** Test of reliability statistics for pilot study.

| Dimensions | Measured Factors (10) | Items | Cronbach's Alpha |
|---|---|---|---|
| Smart Members | Knowledge | 3 | 0.851 |
| | Participation | 3 | 0.919 |
| | Communication | 3 | 0.907 |
| Smart Economy | Collaboration | 4 | 0.908 |
| | Value Proposition | 5 | 0.945 |
| | Value Co-creation | 3 | 0.939 |
| Smart Governance | Democracy | 3 | 0.851 |
| | Regulation | 4 | 0.827 |
| | Monitoring and Evaluation | 5 | 0.934 |
| | Corporate Governance | 5 | 0.961 |

### 3.4.2. Data Collection

The data collection was accomplished through the online survey. The researcher used Google Forms because the use of an online survey enabled access to the respondents. Moreover, this method reduces cost and time, compared to face-to-face distribution, in reaching out to co-operatives' stakeholders (members, co-operative staff, directors, and regulators), which were spread throughout Thailand.

### 3.4.3. Measurement of Variables and Evaluation of Structural Model

The objective of this study was to quantify the extent of the relation between the latent variables (smart members, smart economy, and smart governance). Most statistical methods cannot estimate latent variables. For that reason, only structural equation modeling (SEM) could meet the need of this study because SEM can measure relationships between one or more independent variables and observable variables or latent variables [100], either continuous or discrete, and one or more dependent variables, either continuous or discrete, to be examined [18]. Therefore, in this study we proposed four levels of data analysis: (1) advanced multivariate models; (2) basic multivariate models; (3) intermediate data analysis; and (4) standard deviation or SD; as well as fundamental data analysis. The level of data analysis was selected as follows:

Confirmatory factor analysis (CFA) is a statistical technique used to confirm structural models and explore the structure of a set of observed variables in order to understand

interactions among them [99,130]. Furthermore, CFA is a crucial part of structural equation modeling (SEM) which presents the causal relationships among a dependent variable and independent variables, and it is used in structural analyses for model validation. In this study, the model was assessed using Chi-squared goodness of fit statistics ($p > 0.05$), the comparative fit index (CFI > 0.95), the incremental index of fit (IFI > 0.95), the goodness-of-fit index (GFI > 0.95), the root mean squared error of approximation (RMSEA < 0.05), and normed-fit index (NFI > 0.95) [130–132].

The CFA was tested divided into three steps: measurement model, first-order CFA model, and second-order CFA model. The first step is the measurement model to test the latent variable of smart co-operative management as a smart member, smart economy, and smart governance. The results are in a measurement model which examines the relationship between the latent variables and their observed variables. The second step is the first-order CFA model that examines the relationship between the latent variables in the model. The last step is the second-order CFA model that examines the relationship between constructs to confirm that the analysis results are fit with theorized construct. As a result, the goal of confirmatory factor analysis is to see if the data fit a measurement model that has been proposed [130,132].

The construct validity test captures convergent and discriminant validity to confirm the validity of the measurement model. The crucial assumptions of factor analysis are concepts, rather than statistics [133]. The criteria for construct validity tests are as follows: convergent validity, discriminant validity, and goodness-of-fit tests. The details of the three tests are as follows:

Convergent validity: Three indices were used in this study to evaluate convergence adequacy as follows.

1. Standardized factor loading: It should be statistically significant, and factor loading should be above 0.5 for an acceptable range [133].

2. Average variance extracted (AVE): The AVE is an indicator of convergence; it refers to the mean-variance extracted for the items loading on a construct. The AVE should be above 0.5 [99,133,134]. The AVE is derived from Equation (1) as follows:

$$\text{AVE} = \frac{\sum \lambda^2}{n} \tag{1}$$

Composite reliability (CR): The CR is an indicator of convergence; it is a relatively precise reliability index in SEM models. High construct reliability indicates the existence of internal consistency. A good CR should be 0.7 or higher, although a CR greater than 0.5 is acceptable [99,133]. The CR is derived from Equation (2) as follows:

$$\text{CR} = \frac{\left( \sum \lambda \right)^2}{\left( \sum \lambda \right)^2 + \sum \varepsilon} \tag{2}$$

Discriminant validity: This test provides evidence of the uniqueness of construct and was accepted when: (a) AVE values were significantly higher than squared correlations between constructs in a measurement model [100] and (b) there was no status of cross-loadings between two items in each construct and the SRC was used as an indicator with an acceptable value of more than 4.00 [135]. Adding parameters and removing items were examined only when theoretical and conceptual reasons were specified [136].

Goodness-of-fit tests: The measurement model was assessed by goodness-of-fit (GOF). Alternative GOF measures have been developed for the sensitivity of the $\chi^2$ statistic to sample size, and they were categorized into three groups: absolute measures, incremental measures, and parsimony fit measures, and the authors of [137] suggested that one of each class of measures used to assess a model's goodness-of-fit at a minimum of three to four indices was enough to provide sufficient evidence of model fit.

## 4. Results and Discussion

### 4.1. Result

The following section discusses the key success factors for the smart co-operative, categorized by the dimension of the smart concept [41]. The assessment of the structural model was processed through the SEM approach. In previous studies, many statistical techniques have been used to measure model fit. In this study, we chose important quality indices such as Chi-squared ($\chi^2$), degree of freedom df, $\chi^2$/df, CFI, TLI, IFI, GFI, RMR, and RMSEA to be the goodness-of-fit (GOF) tests' indices. There are minimum requirements to achieve these quality indices in terms of model fitting. The model was assessed by means of Chi-square goodness of fit statistics ($p > 0.05$), comparative fit index (CFI, > 0.95) [138], a ratio of the Chi-squared statistic to the respective degrees of freedom (($\chi^2$/df) < 2 indicates a good model fit) [133], RMSEA and RMR which should be < 0.05 [139], the comparative fit index CFI > 0.95, the goodness-of-fit index (GFI) which should be > 0.95, which is acceptable as a close model fit [100,139,140]. Additionally, the incremental-fit index (IFI) and Tucker–Lewis index (TLI) have also been considered in this study; these indices should be IFI > 0.95 and TLI > 0.95 [139–141]. The research model was accepted with no further modification. The estimation of the parameters was acceptable, and the statistics provided by this study were taken as final values, and they showed that all tests achieved the test requirements. The results of the structural equation modeling (SEM) analysis and the confirmatory factor analysis (CFA) are presented in Table 4. For the smart co-operative, this paper provides three dimensions: member, economy, and governance, which are composed of the smart concepts, co-operative principles, business models, and the framework of IT management. In this study, we used structural equation modeling analysis as a second-order confirmatory factor analysis via Analysis of Moment Structures (AMOS) software, version 22.0.0. The research model was accepted with the following results for the smart co-operative framework, with the relative Chi-square ($\chi^2$) = 28.206, df = 20, $p$ = 0.105, Chi-square ($\chi^2$)/df = 1.410, GFI = 0.987, NFI = 0.995, TLI = 0.997, CFI = 0.999, RMSEA = 0.031, and RMR = 0.006. This model passed the discriminant validity index test, with $\chi^2$/df < 2; RMSEA and RMR < 0.05; and CFI, GFI, AGFI, NFI, and NNFI > 0.95.

**Table 4.** The result of the second-order confirmatory factor analysis on smart co-operative.

| Latent / Observe | Smart Member $\beta_i$ | $b_i$ | S.E. | Smart Economy $\beta_i$ | $b_i$ | S.E. | Smart Governance $\beta_i$ | $b_i$ | S.E. | $r^2$ |
|---|---|---|---|---|---|---|---|---|---|---|
| Knowledge | 0.868 * | 1.000 * | - | | | | | | | 0.754 |
| Participation | 0.888 * | 0.974 * | 0.039 | | | | | | | 0.788 |
| Communication | 0.890 * | 0.936 * | 0.037 | | | | | | | 0.792 |
| Collaboration | | | | 0.958 * | 0.966 * | 0.025 | | | | 0.917 |
| Value Proposition | | | | 0.930 * | 0.894 * | 0.023 | | | | 0.866 |
| Value Co-creation | | | | 0.921 * | 1.000 * | - | | | | 0.848 |
| Democracy | | | | | | | 0.844 * | 0.932 * | 0.029 | 0.712 |
| Regulation | | | | | | | 0.923 * | 1.000 * | - | 0.851 |
| Monitoring and Evaluation | | | | | | | 0.899 * | 0.873 * | 0.029 | 0.808 |
| Corporate Governance | | | | | | | 0.865 * | 0.764 * | 0.026 | 0.749 |

| Latent | Smart Co-Operative $\beta_i$ | $b_i$ | S.E. | $R^2$ |
|---|---|---|---|---|
| Smart Member | 0.954 * | 0.877 * | 0.012 | 0.910 |
| Smart Economy | 1.000 * | 1.000 * | - | 1.000 |
| Smart Governance | 0.966 * | 0.970 * | 0.013 | 0.934 |

$\chi^2$ = 28.206, df = 20, Relative $\chi^2$ = 1.410, $p$-Value = 0.105, GFI = 0.987, NFI = 0.995, TLI = 0.997, CFI = 0.999, RMSEA = 0.031, RMR = 0.006

Note: * $p < 0.05$.

The estimation of the parameters was acceptable, showing that all tests achieved the test requirements. Table 4 presents a summary of the model fit and quality index results. Therefore, the smart co-operative framework consists of three variables: smart members, the smart economy, and smart governance. Figure 4 indicates the standardized coefficient and factor loading results for the second order of the smart co-operative dimensions, which loads quite strongly. The analysis result also shows that the smart economy dimension was the most significant impact of all the three factors on the smart co-operative, followed by the smart governance dimension and smart members (standardized regression weights 1.000, 0.966, and 0.954). The details of the structural equation modeling analysis as second-order confirmatory factor analysis (CFA) of the smart co-operative framework are as follows.

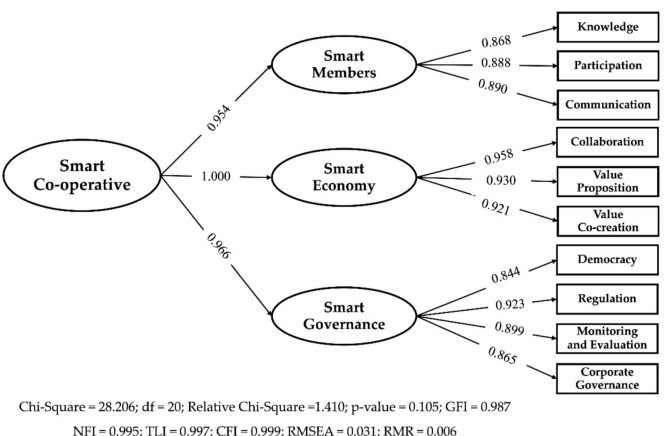

Chi-Square = 28.206; df = 20; Relative Chi-Square =1.410; p-value = 0.105; GFI = 0.987
NFI = 0.995; TLI = 0.997; CFI = 0.999; RMSEA = 0.031; RMR = 0.006

**Figure 4.** Smart co-operative conceptual model.

The first dimension is the smart member. There are three factors in the smart member dimension. Communication had the most significant impact of all three on the smart member dimension, followed by participation and knowledge (standardized regression weights 0.890, 0.888, and 0.868).

The second dimension is the smart economy. There are three factors in the smart economy dimension. Collaboration had the most significant impact of all three on the smart economy followed by value proposition and value co-creation (standardized regression weights 0.958, 0.930, and 0.921).

The last dimension is the smart governance. There are four factors in the smart governance dimension. The regulation was the most significant impact of all four on smart governance impact followed by monitoring and evaluation, corporate governance, and democracy (standardized regression weights 0.923, 0.899, 0.865 and 0.844).

### 4.2. Discussion

In this section, we discuss and present a smart co-operative conceptual model based on the findings from the Results section as showing the main contributions of this research in investigating the effect of the critical factors of the smart co-operative—the need to improve co-operative services and management is a major challenge in effectively guiding the sustainable development of co-operatives. It is necessary to integrate co-operative principles, business models, and ICT to promote co-operative principles, co-operative business, and co-operative governance in smart co-operatives. Presently, ICT plays highly critical roles in facilitating this process. In co-operatives, it was found to be the main objective and achievement in relation to co-operative business. Based on the evidence, this study proposes the smart co-operative concept, with its key factors divided into three dimensions and integrated with ICT to promote and support all three smart co-operative dimensions: smart member, smart economy, and smart governance. Figure 3 shows the proposed model.

The dimension of smart members encourages the role of members. ICT has a role in supporting knowledge management, participation, and communication among members. Knowl-

edge is necessary for improving the services and performance of co-operatives [18,45,47] in order to ensure financially efficient and effective sustainable development of co-operatives [9,46]. Additionally, the participation of members is crucial for the process of devising and improving decision-making processes in various activities that affect co-operatives [27,39,44,45]. Additionally, members should interact and communicate with one another in order to exchange information and provide information to policymakers [47,53]. Communication plays an important role in creating awareness and understanding between the members and the administrations [142].

The dimension of the smart economy is covered in our framework in order to promote co-operative business and services. ICT plays a crucial role in supporting collaboration, value proposition, and value co-creation for members. These are effective for co-operative services and management quality. Based on the literature, the collaboration of diverse stakeholders in executing policies and action plans can ensure the quality and improvement in the decision-making process [27,31,38,88]. Value co-creation in co-operative business, services, digital business models can lead to the creation of value and value proposition and can deliver value for both customers and other stakeholders [57,107–109]. The value proposition concept should consider all the stakeholders' needs in order to meet members' needs and achieve a competitive advantage [110–113]. Additionally, ICT-enabled services improve the efficiency and enhance quality of services, efficiently deliver services, and increase the quality of member life [53,85].

The dimension of smart governance is covered in our framework to encourage and promote the role of governance. ICT plays an important role in supporting governance processes. In previous studies, co-operatives were controlled by means of a democratic system to provide benefits to their members [2–4,10]. The operation of a co-operative has an impact on various stakeholders. Therefore, regulation, monitoring and evaluation, as well as corporate governance are essential factors to create transparency and trust in co-operatives. In addition, smart governance involves using ICT to create new channels for governance, such as e-governance and e-democracy, with processes such as e-voting [53,71,72] to support democratic decision-making processes and enhance transparency in governance [11,32,65,73]. The participation of stakeholders in monitoring the management of the co-operative can improve decision-making processes [27,39,45]. A governance mechanism involves monitoring and controlling the quality of information to improve and promote effective management and prevent the occurrence of all risk cases [5,82,83]. Corporate governance is an influential driver which contributes to ownership feelings and to success in terms of engaging with and motivating members [11]. Furthermore, ICT has an important role in facilitating data quality, information sharing, integration, and monitoring and in significantly enhancing accuracy and the effectiveness of decision making [22,61,63,76]. Moreover, another key role of smart governance is IT governance, a term referring to the use of technologies to achieve governance processes that are necessary for effectiveness, efficiency, and increasing transparency [52,91,96,97]. Hence, smart governance, through the use of information and communication technologies (ICT) for governing, helps to improve monitoring, controlling process, decision making, and direct democracy and to increase transparency in the co-operative sector [19].

## 5. Conclusions

In this study we have presented the results of a questionnaire which was distributed to co-operative sector stakeholders in Thailand in order to understand and to investigate the critical success factors (CSFs) that are effective in the smart co-operative management framework. This research model was developed, and ten factors were formulated in this study to contribute to the conceptual framework of smart co-operatives for the purpose of co-operative management. The fundamental concept of this proposed model has been divided into three dimensions, with the identification of ten key principles. Firstly, the smart member category consists of three key principles, which are (1) knowledge, (2) participation, and (3) communication. Secondly, the smart economy dimension consists of three key

principles, which are (1) collaboration, (2) value proposition, and (3) value co-creation. Finally, smart governance consists of four key principles which are (1) democracy, (2) regulation, (3) monitoring and evaluation, and (4) corporate governance. Co-operative business processes and business models have recently changed. As a result, ICT has played an important role in co-operative organization. Moreover, the co-operative business value has grown, and the performance of co-operatives has revealed governance and management problems [18]. Therefore, the research model of this study was evaluate using the structural equation modelling (SEM) approach. To use SEM to evaluate the research model and the determining factors of the smart co-operative framework to enhance the quality and efficiency of co-operative services and management, the structural model was investigated using confirmatory factor analysis (CFA) and model fit. Regarding confirmatory factor analysis (CFA), these results indicated that the ten factors support the model structure, and the model fit achieved the acceptable ranges, passing tests and validating the model's goodness of fit: with CFI, IFI, TLI, RMSEA, and SRMR, there was a probable good fit [142].

The dimension of smart members consists of knowledge, participation, and communication among the members, associated with the significant influence of ICT on the roles of the member of the smart co-operative. Additionally, another main dimension is the smart economy, which consists of collaboration, value proposition, and value co-creation of the members, again associated with the significant influence of ICT on co-operative services and management. The last dimension is smart governance which consists of democracy, regulation, monitoring and evaluation, and corporate governance associated with the significant influence of ICT. Therefore, the smart co-operative involves the solution of integrating new technologies and, for co-operative services, ensuring co-operative management. The results of this study indicate that smart members, smart economy, and smart governance are crucial in smart co-operatives. The study shows that all factors have an impact on co-operative process especially in the areas of service, management, and governance. The smart co-operative model would be a useful tool for promoting sustainable development with efficient management, as well as transparency and trust in co-operatives.

**Author Contributions:** Conceptualization, A.C.; methodology, A.C.; validation, S.K.; formal analysis, A.C.; investigation, A.C. and S.K.; resources, A.C.; data curation, A.C.; writing—original draft preparation, A.C.; writing—review and editing, A.C. and S.K.; visualization, A.C.; supervision, S.K. All authors have read and agreed to the published version of the manuscript.

**Funding:** This research was funded by the Agricultural Research Development Agency (Public Organization). ARDA is a public government organization under the supervision of the Minister of Agriculture and Cooperatives of Thailand.

**Institutional Review Board Statement:** The study was conducted according to the guidelines of the Declaration of Helsinki, and approved by the Institutional Review Board (or Ethics Committee) of Mahidol University (Protocol code COE NO. MU-CIRB 2022/011.3101 and date of approval 31 January 2022).

**Informed Consent Statement:** Informed consent was obtained from all subjects involved in the study.

**Data Availability Statement:** No new data were created or analyzed in this study. Data sharing is not applicable to this article.

**Acknowledgments:** The authors would like to thank the anonymous experts and reviewers for their valuable comments and suggestions that have improved this article.

**Conflicts of Interest:** The authors declare no conflict of interest.

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
