# Peer review of "Sustainable Development: Smart Co-Operative Management Framework"

_sustainability, doi:10.3390/su14063641_

Round 1

Reviewer 1 Report

Co-operatives are financial institutions created by a group of people working together in the same location/nation/region/occupation to provide credits, goods, and services to members.

This study presents a conceptual model for smart cooperatives. The authors  evaluate/validate the proposed model with a panel of experts, stakeholders in the cooperative sector in Thailand using a semi-structured questionnaire.

The study proposes the smart co-operative model integrated with ICT to promote and support three smart co-operative dimensions: smart member, smart economy, and smart governance. 

The key result is that the smart co-operative model is shown to be a useful tool to promote sustainable development, efficient management, transparency and trust.

The paper is extensive and rich in detail. It gives a thorough overview of the theoretical framework and the methodology applied. The authors clearly spell out their theoretical assumptions, methodological apparatus, as well as the results and contributions of the model.

A few print mistakes, e.g., 

Line 40  Co-operatives differ

Line 794 that are effective

Author Response

Dear Editor and Reviewer of MDPI Sustainability

            Thank you for the opportunity to revise and improve our manuscript. We really appreciate your valuable comments and suggestions. We expect our manuscript has improved upon the suggested revision.

All the detailed responses: "Please see the attachment"

Please do not hesitate to contact us if you have any comments or suggestions regarding the revision of our manuscript. We believe that our manuscript would be great upon your comments and suggestions. We look forward to hearing from you soon.

Sincerely, 

Supaporn Kiattisin

Reviewer 2 Report

The paper deals with important and interesting topics.

It is well written but some improvements are needed.

Please avoid using references in the abstract.

Why did you write www.ica.coop as a reference and not like it is given in the instructions for authors at the very beginning of the article?

Please introduce research questions and hypotheses in the introduction. In addition, at the end of the introduction chapter explain the organisation of your paper.

Chapter 2 should be renamed into Literature review.

Some figures are barely readable. Please provide figures in higher resolutions.

Similarly, some parts of texts are difficult to read, follow and understand. For example, the whole chapter 2.1.3. is written as one paragraph which is one and half page long. Please introduce a more logical paragraph to make the article easier to follow.

The parts related to survey and sample descriptions have to be improved.

The sentence "The sample size is determined based on types of statistical power analyses, and effect size [130], which is structural equation modeling (SEM)." does not make any sense. SEM does not have anything with the determination of the sample size. You used power analysis and effect size for that. However, it is unclear how you did that. No additional information about conducted sampling is provided.

Which sampling procedure did you apply? What is the population exactly? What is the population size? What was your sampling frame?

I am curious - how did you get to the conclusion that "... 431 samples were required to be collected for this study." I would say that you concluded that the sample size should be 431 observed units (which is unclear how you make this conclusion).

Please provide more information about your questionnaire.

Used statistical methods should be briefly introduced and explained.

Author Response

Dear Editor and Reviewer of MDPI Sustainability

            Thank you for the opportunity to revise and improve our manuscript. We really appreciate your valuable comments and suggestions. We expect our manuscript has improved upon the suggested revision. A  point-by-point response to the reviewer’s comments: Please see the attachment.

Please do not hesitate to contact us if you have any comments or suggestions regarding the revision of our manuscript. We believe that our manuscript would be great upon your comments and suggestions. We look forward to hearing from you soon.

Sincerely, 

Supaporn Kiattisin
